# Feeding and Growth Response of Fall Armyworm ***Spodoptera frugiperda* (Lepidoptera: Noctuidae)** towards Different Host Plants

**DOI:** 10.3390/insects15100789

**Published:** 2024-10-10

**Authors:** Muhammad Saqib Ajmal, Sajjad Ali, Aftab Jamal, Muhammad Farhan Saeed, Emanuele Radicetti, Stefano Civolani

**Affiliations:** 1Department of Entomology, Faculty of Agriculture and Environment, The Islamia University of Bahawalpur, Bahawalpur 63100, Pakistan; saqibajmalentomologist@gmail.com; 2Department of Soil and Environmental Sciences, Faculty of Crop Production Sciences, The University of Agriculture, Peshawar 25130, Pakistan; aftabses98@gmail.com; 3Department of Environmental Sciences, COMSATS University Islamabad, Vehari Campus, Vehari 61100, Pakistan; farhansaeed@cuivehari.edu.pk; 4Department of Chemical, Pharmaceutical and Agricultural Sciences, University of Ferrara, Via Borsari 46, 44121 Ferrara, Italy; rdcmnl@unife.it

**Keywords:** feeding indices, larval growth, nutritional compositions, plant minerals

## Abstract

**Simple Summary:**

The fall armyworm, *Spodoptera frugiperda*, is a polyphagous insect pest. The host plants influence the feeding of herbivorous insects, which affects their growth and development. To determine the effect of the nutrient and mineral contents of different host plants (castor bean, cotton, maize, okra, cabbage, and sugarcane) on the growth and development of *S. frugiperda*, feeding indices and biological parameters were calculated. The tested host plants significantly differed in their nutritional and mineral chemistry. The feeding indices on these host plants showed a significant difference. In the present study, the maximum larval growth, pupal length, pupal weight, and feeding indices were recorded in *S. frugiperda* larvae that fed on maize and castor bean leaves. It indicated that suitable nutrients were found in maize and castor bean plants for *S. frugiperda*. The outcome can help in devising effective management for *S. frugiperda*.

**Abstract:**

The fall armyworm, *Spodoptera frugiperda*, is a major migratory polyphagous insect pest of various crops. The essential nutrient and mineral profile of the host plants determines the feeding fitness of herbivorous insects. As a result, the growth and development of insects is affected. To determine the effect of the nutrient and mineral profile of different host plants (maize, castor bean, cotton, cabbage, okra, and sugarcane) on the growth and development of *S. frugiperda*, biological parameters like larval weight, pupal weight (male/female), and feeding and growth indices were calculated. The proximate compositions such as crude protein, crude fat, crude fibre, and ash and mineral contents of the tested host plants showed significant differences (*p* < 0.05). The feeding indices on these host plants also differed significantly (*p* < 0.05). The maximum relative growth rate (RGR), relative consumption rate (RCR), and consumption index (CI) were recorded in *S. frugiperda* larvae that fed on maize and castor bean leaves. The crude protein, dry matter, and ash contents in maize and castor bean were significantly higher and positively correlated with the RGR and RCR of *S. frugiperda* larvae. The larval, male and female pupal weights were the maximum in the larvae feeding on the castor bean host plant. These findings provide novel information based on nutritional ecology to develop sustainable integrated pest management strategies using selective crop rotation.

## 1. Introduction

Insect herbivores confront numerous constraints when they consume their host plants [1]. Plants have evolved certain physical barriers (thick leaves and glandular trichomes) and a wide range of chemical defences to cope with insect herbivores [2,3,4]. A diverse range of natural compounds and constituents, which are unique to a certain plant, act as deterrents, repellents, or toxins and impede the digestion of food. However, insect herbivores have evolved mechanisms to counteract these defences, by escaping detection or suppressing plant defences [5]. Polyphagous insect herbivores compete with a wide range of plant defences from various taxa, but their metabolic system is less optimized than that of mono- and oligophagous herbivores [6,7,8].

Minerals and nutrients are essential for the insects to grow, develop, maintain tissues, reproduce, and to obtain energy. Plants with higher mineral contents are more resistant to insect pests because they are stronger and healthier, and these minerals may alter the suitability of the host plants for the insect herbivores [9]. Mineral contents in plants, such as phosphorus, potassium, and calcium, may alter insect physiology and morphology [10,11]. Additionally, the role of plant fat, fibre, and protein contents in the growth and development of insect herbivores cannot be overlooked [12]. Plants with a higher protein content may promote the growth and development of herbivorous insects by increasing their body size, shortening life cycles, and increasing fecundity [13]. Therefore, polyphagous insect herbivores have a benefit from better nutritional balance, increased resource availability, and ability to dilute the particular host plant’s defences by consuming a variety of plant species; however, their growth and survival and population dynamics may vary on different plant species having varying nutritional and mineral contents [14,15,16].

The nutritional indices and development of insects are altered by consuming different host plants with varied nutritional contents. In addition to host plant defence mechanisms, the nutritional and mineral profiles of the host plants can also be helpful in devising pest management strategies by altering the host plant preference, lifecycle, and biology of the insect herbivores [17,18]. The host plants can be managed under crop selection and rotation plans. While plant nutrient and mineral constituents may be managed with good crop breeding and soil nutrition. Crop varieties with higher nutrients use efficiency and enhanced pest resistance are helpful for this purpose [19,20]. This can be attained through balanced fertilization based on soil tests and using nutrient and mineral modifications to reinforce the natural host plant defences. Additionally, crop rotation and intercropping practices are also critical for disrupting the insect pest life cycle and maintaining the soil health [21]. So, the investigation of nutritional indices and insect growth can help to understand the physiological and behavioural basis of the insect herbivores’ response towards different host plants.

The fall armyworm (FAW), *Spodoptera frugiperda* J.E. Smith (Lepidoptera: Noctuidae), is a major polyphagous pest that can rapaciously diminish the production of several agricultural crops [22]. It has high fecundity, potential for migration, and capacity to develop resistance against a variety of insecticides [22,23]. It feeds on more than 350 plant species from 76 families, such as wheat, barley, sorghum, maize, and soybean [24]. Recently, the FAW has posed a significant threat to maize production worldwide. Maize is a very important cereal crop in the world and stands in third place after wheat and rice based on its production [25]. It is used for human, poultry, and livestock both as food and feed. Now, the FAW is becoming a major threat to maize and other crops in Pakistan resulting in massive crop losses [26].

Focusing on the economic importance of *S. frugiperda*, the current study was designed with the goal of determining the effect of the mineral and nutritional constituents of different host plants including castor bean (*Ricinus communis* L.), cotton (*Gossypium hirsutum* L.), maize (*Zea mays* L.), cabbage (*Brassica oleracea*), okra (*Abelmoschus esculentus* L.), and sugarcane (*Saccharum officinarum* L.) on the feeding indices, growth, survival, and development of *S. frugiperda*.

## 2. Materials and Methods

### 2.1. Study Area

The present study was conducted at the Department of Entomology, Faculty of Agriculture and Environment, The Islamia University of Bahawalpur, Pakistan. The city is located in a plain area at an altitude of 214 m above sea level, with a longitude of 29°25′5.0448′′ N and a latitude of 71°40′14.4660′′ E.

### 2.2. Insect Culture

*Spodoptera frugiperda* larvae were collected from insecticide-free maize plants in university’s research field area. The larvae were placed in cylindrical plastic screened cages (30 cm h, 18 cm d) and allowed to feed on caster bean leaves under laboratory conditions (25 ± 2 °C, R.H 65 ± 5%, and 16 L:8 D photoperiod) [27]. After pupation, the male and female pupae were identified, paired, and placed in Petri dishes and moved to cylindrical plastic screened cages for mating and egg production after adult emergence. Egg laying sheets were hung from the top into the rearing cages as oviposition substrate and were fixed with rubber bands. The newly emerged FAW adults were provided with a 20% honey solution on cotton plugs. After egg laying, the sheets were removed and shifted into new, screened plastic cages until hatching. The larvae were reared on caster bean leaves under the same laboratory conditions until the next generation. Third-instar *S. frugiperda* larvae from the 6^th^ generation were used for the experiments.

### 2.3. Host Plants

Different host plants including castor bean (*Ricinus communis* L.), cotton (*Gossypium hirsutum* L.), maize (*Zea mays* L.), okra (*Abelmoschus esculentus* L.), cabbage (*Brassica oleracea*), and sugarcane (*Saccharum officinarum* L.) were maintained as insecticide-free sources of feed for *S. frugiperda* larvae at the field research area of the Islamia University of Bahawalpur. Fresh leaves of these host plants were used for the experiments.

### 2.4. Analyses of Nutritional and Mineral Profiles of Host Plants

The nutritional and mineral profiles of the host plants were analysed at the Department of Animal Nutrition and Central Laboratory Complex, University of Veterinary and Animal Sciences, Lahore, respectively by using standard protocols (AOAC, 2003). Fresh plant leaves were taken, washed with distilled water, and oven dried at 70 °C to a constant weight for 24 h to determine dry matter and then ground into a fine powder with a blender. Five grams of dried leaves were used to calculate the crude ash, crude protein, crude fibre, crude fat, dry matter, and mineral contents of the plants. Each sample was replicated three times from three plants of same species. The crude protein was determined by Kjeldahl analyses. Crude fat was analysed by exhaustive Soxhlet extraction using petroleum ether (40–60 °C, BP), and crude fibre was estimated using a fibre analyser. Atomic absorption spectrophotometer (STA−4800 Spectrophotometer, Stalwart Analytics) was used to determine the mineral contents of the host plants including calcium (Ca), potassium (K), magnesium (Mg), phosphorus (P), and zinc (Zn).

### 2.5. Experimentation

The experiment was conducted in a growth chamber (6 × 6 feet) with a controlled temperature, light period, and humidity. Three hundred newly hatched first-instar larvae were randomly divided into six groups and were allowed to feed on six different host plants, i.e., castor bean, cotton, maize, okra, cabbage, and sugarcane in screened plastic containers (32 cm × 15 cm × 15 cm). After the 3^rd^ instar, they were shifted individually into sealed plastic Petri dishes (8 cm in diameter). The Petri dishes were provided with wet filter papers in the bottom to maintain humidity and avoid water loss from plant leaves [28]. Quantified, fresh host plant leaves (1000 mg/larva) were offered to the larvae daily until they pupated. Their survival was checked on a daily basis. The pupae were separated on a gender basis. The newly emerged adults were paired (female and male) in screened plastic containers (32 cm × 15 cm × 15 cm) and nourished with a 20% honey solution provided through cotton plugs. Egg sheets (baby liner cloth), used as oviposition substrates, were hung from the top of the containers, and fixed with strong rubber bands. The eggs were collected daily and placed in plastic Petri dishes (8 cm diameter) until hatching. Each treatment was replicated five times. The effects of different plant feeds on biological parameters in terms of larval and pupal weights, larval and pupal lengths, and feeding indices were calculated during the whole experiment. The larval weight on the 1^st^ day of each instar and the pupal weight on 2^nd^ day were obtained by a high-precision electronic weighing balance (ATX/ATY Unibloc Analytical Balance, Shimadzu Scientific Instruments, Kyoto, Japan). The larval and pupal lengths were measured by using a precision stainless-steel scale (Shinwa 150 mm Rigid, Model 13005, Shinwa Measuring Tools Corporation, Schaumburg, IL, USA). The larvae were shortly exposed to CO_2_ to make them immobile, and their body was slightly relaxed on a filter paper to measure their lengths.

### 2.6. Feeding Indices

The weights (mg) of larvae, plant leaves, and faeces were recorded at an interval of 24 h to calculate the feeding indices by using a high-precision balance (ATX/ATY Unibloc Analytical Balance, Shimadzu Scientific Instruments, Kyoto, Japan) with a range of 200 g/0.0001 g. The feeding indices were calculated from the following equations [29]:$$Relative\;Growth\;Rate\;\left(RGR\right)=\frac{\Delta B}{B}\times T$$
$$Relative\;Consumption\;Rate\;\left(RCR\right)=\frac{D}{B}\times T$$
$$Efficiency\;of\;conversion\;of\;ingested\;food\;\left(ECI\right)=\frac{B}{D}\times 100$$
$$Consumption\;Index\;\left(CI\right)=\frac{D}{F}\times 100$$

Here, *ΔB* is the larval weight gain during the feeding period; *B* is the initial mean larval weight; *D* is the food biomass ingested per larva; *F* is the faeces biomass produced per larva; and *T* is the feeding period (days).

### 2.7. Statistical Data Analyses

Data regarding the larval weight and length, pupal weight and length, larval survival, and feeding indices (RGR, RCR, ECI, and CI) were subjected to a factorial ANOVA, with means compared using the Duncan multiple range (DMR) test at a 5% significance level. Values for the nutritional contents (crude ash, crude protein, crude fat, dry matter, and crude fibre) and mineral contents (Ca, Mg, K, P, and Zn) from the host plants were subjected to a one-way ANOVA, with means compared using the DMR test at a 5% significance level. To meet the assumption of normality, the data for mineral contents and feeding indices were transformed using the square-root formula [SQRT (x + 0.5)] [30]. Levene’s test was applied to assess the assumption of homogeneity of variances for all the ANOVA analyses. In cases in which heteroscedasticity (inequality of variances) was detected, Welch’s ANOVA was employed as an alternative. Post hoc comparisons were performed using DMR tests following both standard ANOVA and Welch’s ANOVA where appropriate. The feeding indices were correlated with the proximate composition analyses using Pearson correlation. The tables for the mineral composition and feeding indices present the original mean (±SE) values. All these analyses were performed using the SPSS software, version 21 (SPSS Inc., Chicago, IL, USA).

## 3. Results

### 3.1. Proximate Composition of Different Host Plants Used to Feed S. frugiperda Larvae

The homogeneity of variances confirmed the hypothesis on equal variances for all the tested variables (*p* > 0.05). However, for dry matter, the homogeneity of variances approached the threshold (*p* = 0.062), whereas the *p*-values from the Welch’s ANOVA are below 0.05. The tested host plants differed significantly (*p* < 0.001) in their proximate nutritional composition in terms of ash (F (5, 12) = 110.7, *p* < 0.001), crude protein (F (5, 12) = 1149.4, *p* < 0.001), crude fibre (F (5, 12) = 1742.8, *p* < 0.001), dry matter (F (5, 12) = 1154.3, *p* < 0.001), and crude fat (F (5, 12) = 292.9, *p* < 0.001). The maximum ash content (12.13%) was found in the maize plant sample, whiles the minimum ash content (6.67%) was detected in the sugarcane plant sample. The crude protein content was the highest (23.48%) in the castor bean plant, and sugarcane contained the lowest (4.57%) crude protein value. The maximum (33.20%) crude fibre value was recorded in the case of the sugarcane plant, while its minimum (9.36%) value was observed in the maize sample. Similarly, the highest dry matter (43.65%) was recorded in the castor bean plant sample, and the lowest (15.80%) dry matter content was found in the cabbage plant sample. The maximum crude fat (9.50%) was found in the cotton plant sample, while its minimum (0.87%) content was found in the cabbage plant sample (Table 1).

### 3.2. Mineral Composition Analysis of Different Host Plants Used to Feed S. frugiperda Larvae

The homogeneity of variances suggests that the assumption of equal variances is accepted for most of the variables; however, a slight deviation in zinc content was observed. The *p*-values from the Welch’s ANOVA were below 0.05, indicating that the mineral contents varied significantly between the tested host plants. These host plants showed a significant difference (*p* < 0.001) among their tested mineral contents including calcium (Ca) (F (5, 12) = 2323.1, *p* < 0.001), potassium (K) (F (5, 12) = 190.8, *p* < 0.001), magnesium (Mg) (F (5, 12) = 2456.8, *p* < 0.001), phosphorus (P) (F (5, 12) = 99.5, *p* < 0.001), and zinc (Zn) (F (5, 12) = 4638.2, *p* < 0.001). The maximum Ca content (28.33 mg) was found in the okra plant sample, while the minimum Ca (2.49 mg) was found in the cotton plant. The highest Mg contents (38.92 mg) were detected in the cabbage plant sample, while the lowest (1.36 mg) Mg contents were present in the sugarcane plant. The maximum (26.31 mg) K content was found in the cabbage plant, while its minimum (15.93 mg) content was seen in the cotton plant. Additionally, the highest Zn contents (0.82 mg) were recorded in the case of the okra plant sample, while its lowest (0.20 mg) contents were found in the castor bean plant sample (Table 2).

### 3.3. Feeding Index Parameters for S. frugiperda Larvae Feeding on Different Host Plants

The homogeneity of variance shows that hypothesis for equal variances for feeding indices is accepted (*p* > 0.05) with a slight variation in the relative growth rate (RGR) data. The feeding index calculations revealed that the tested host plants including castor bean, cotton, maize, okra, cabbage, and sugarcane significantly affected the relative growth rate (RGR) (F (5, 20) = 41.5, *p* < 0.001), relative consumption rate (RCR) (F (5, 20) = 486.6, *p* < 0.001), and consumption index (CI) (F (5, 20) = 3.14, *p* < 0.05) of the *S. frugiperda* larvae; however, the conversion of ingested food (ECI) (F (5, 20) = 2.61, *p* > 0.05) was non-significant. The highest RGR (1.13 mg/mg/day) for the *S. frugiperda* larvae was observed when they fed on maize, while the lowest RGR (0.59 mg/mg/day) was recorded when they fed on sugarcane. The *S. frugiperda* larvae also exhibited a similar trend in the case of the RCR and CI calculations, in which the maximum RCR (50.47 mg/mg/day) and CI (68.39%) values were calculated when the *S. frugiperda* larvae fed on maize. The highest ECI value (71.80%) was recorded for castor bean. The *S. frugiperda* larvae showed the minimum RCR and CI (19.2 mg/mg/day and 65.69%, respectively) values when they fed on sugarcane (Table 3).

### 3.4. Growth Parameters for S. frugiperda Larvae Feeding on Different Host Plants

#### 3.4.1. Larval Length and Weight

From the data, it was depicted that feeding on different host plants significantly affected the length (F (5, 54) = 1860, *p* < 0.001) and weight (F (5, 54) = 1002, *p* < 0.001) of the *S. frugiperda* larvae. The highest larval length (32.97 mm) and weight (205 mg) were recorded when the *S. frugiperda* larvae fed on castor bean, while the smallest larval length (26.65 mm) and minimum larval weight (140 mg) was recorded when *S. frugiperda* larvae fed on sugarcane (Figure 1).

#### 3.4.2. Pupal Length and Weight

There was a significant difference in the female and male pupal lengths of *S. frugiperda* (F (1, 108) = 828, *p* < 0.001) when they fed on six different host plants (F (5, 108) = 314.16, *p* < 0.001) (Figure 2). The female and male pupae were significantly longer (17.97 and 17.67 mm, respectively) when the *S. frugiperda* larvae fed on castor bean, while the smallest pupal lengths (14.79 and 14.39 mm) were recorded on sugarcane. The *S. frugiperda* female and male pupae also showed significant differences (F (1, 108) = 156.42, *p* < 0.001) in their weights when they fed on six different host plants (F (5, 108) = 2614.15, *p* < 0.001) (Figure 3). The female and male pupae were significantly heavier (198.6 and 191.6 mg, respectively) when the *S. frugiperda* larvae fed on castor bean, while the lowest pupal weights (male 131.8 mg and female 136.2 mg) were recorded when they fed on sugarcane. The female pupae were heavier than the male pupae regardless of the provided plant leaves (*p* < 0.05).

#### 3.4.3. Total Larval Survival Rate

The larval survival of *S. frugiperda* showed a significant difference (F (5, 12) = 951, *p* < 0.001) when the larvae were fed on six different host plants (Figure 4). The highest and statistically similar larval survival rate was revealed on maize, castor bean, and cotton (96.6%, 96.4%, and 96.1%, respectively), while the minimum larval survival rate (62.1%) was seen on sugarcane.

### 3.5. Matrix of Pearson’s Correlation Coefficient among Proximate Analysis with Feeding Indices of S. frugiperda

The results of Pearson’s correlation among the feeding indices of the *S. frugiperda* larvae and the proximate nutritional contents of the host plants are given in Table 4. The RGR had a statistically positive relationship with the RCR, DM, and ash contents. The RCR had a significant positive relationship with the DM and ash content. The DM had a significant positive relation with crude protein. The CP had a significant relation with ash and a negative relation with crude fibre. It means that material higher in protein will definitely produce a higher level of ash and dry matter. A food material rich in crude fibre will be a weaker source of CP, EE, and ash. Statistically, a negative relationship exists between CF and EE. It means that a higher fat content does not warrant better larval growth (Table 4).

## 4. Discussion

In our study, the nutritional and mineral composition of the tested plants differed significantly. These variations might affect the plant–insect interactions, which ultimately affect the spread and population dynamics of the insects on different host crop plants [31]. The nutritional and chemical elements of the host plants are influential in their resistance and tolerance against insect pests [32]. Feeding indices, for instance RGR, RCR, and CI, are vital indicators for recognizing resistance in selected crops and executing pest management tactics. In this study, *S. frugiperda* exhibited the highest feeding indices on maize and castor bean plants. The RCR is linked with food bioavailability, nutrients, minerals, and allelochemicals and affects the growth and development of *S. frugiperda* through nutrient assimilation and conversion [33]. The crude protein had a positive relationship with growth indices. Maize and castor bean were identified as suitable hosts for *S. frugiperda*, supporting higher rates of survival, growth, and development due to their nutritional profiles, particularly their higher protein contents [34,35]. Our findings align with previous studies by Cock et al. [36], Ganiger et al. [37], and Sharanabasappa et al. [38], which reported maize as a preferred host for *S. frugiperda*. The higher protein and fat content and lower fibre content in maize and castor bean may enhance *S. frugiperda* growth and development.

The mineral contents in host plants, like K and P, can improve resistance against insect pests by promoting secondary metabolic compound production and reducing carbohydrate accumulation [9,31,39]. This beneficial effect of minerals is largely predominated in many host plants against plant hoppers, beetles, lepidopterans, and mites [9]. Our findings indicated that maize had a relatively lower mineral content, making it the most suitable host plant for *S. frugiperda*. Incorporating minerals into crops can improve their resistance to pests, as the abridged larval survival and body weight of rice leaf folder and sugarcane borer was evidenced due to higher K levels [9,39]. This highlights the importance of understanding the mineral contents of host plants in devising effective pest management policies.

Insect body growth is influenced by the consumption, utilization, and assimilation of plant food. Protein, nitrogen, carbohydrates, and water contents are the essential primary nutrients for insect growth and development [40,41]. Our findings showed that larval growth and survival was maximum on maize. Altaf et al. [42] reported better growth of *S. frugiperda* on maize compared to sorghum and wheat, likely because of differences in the nutritional and mineral contents and defensive compounds in these plants. These factors influence the host preference, insect survival, and development. Our results align with those of Wang et al. [43], who also found that *S. frugiperda* had the shortest larval development period, longer adult longevity, the highest pupal weight, and the maximum fecundity on maize plants as compared with those in soybean, tomato, and cotton crops. Our results were also in consistent with those of Barros et al. [44] and Ramos et al. [45], who reported that the *S. frugiperda* showed a higher preference for oviposition, development, and reproduction on maize than on millet, cotton, and soybean crops. Host plant suitability is indicated by the factors like larval development, growth, fecundity, and overall generation time, with higher feeding and growth indices [43].

The results showed that *S. frugiperda* had the highest larval and pupal lengths, weights, and survival rates when they fed on maize and castor bean leaves. The female pupae were heavier than the male pupae regardless of the larval host plant. Differences in nutritional quality, chemical stress, and secondary metabolites in these host plants affected the larval and pupal growth, development, and survival [46]. Previously, Awmack and Leather [40] and Kumar et al. [47] reported that the larval and pupal weights of *S. frugiperda* and *Lymantria dispar* (spongy moth) increased when they fed on high-protein diets. Our results are aligned with those of Xie et al. [48], Liu et al. [49], and Marri et al. [50], who observed that the female pupae were heavier than the male pupae. Previously, it was reported that sexual size dimorphism is linked with sexual dimorphic growth time and rate during the larval development because larval instars are longer in females with higher allometric growth rates in most of the insect species [51,52]. Moreover, females invest more in their bodies because of their reproductive biology, which makes them larger than males [53]. The rate of development and survival is directly associated with the quantity and quality of ingested food; as the quantity of ingested food decreases, the insect becomes smaller and lighter in weight with delayed development [54]. The larval and pupal weights and survival of *S. frugiperda* are greatly affected by the host plants, with significantly heavier larvae and pupae recorded on maize. Wang et al. [43] found no significant variation in the male and female pupal weights on the same host plant, while Chen et al. [55] and He et al. [56] found heavier male pupae. These disparities are due to different host plants with varied nutritional and mineral values, which influence the larval fitness, development, and ultimate survival [57]. Moreover, different plant species’ leaves may exhibit variable rates of water loss because of differences in their morphology, cuticle thickness, and stomatal density [58,59]. These features may have a slight effect on the water retention in the plant leaves. That is why, we took necessary measures to minimize the water loss, such as preserving high humidity levels by providing wet filter papers, air-tight Petri plates, and constant running of humidifiers and air conditioner in the controlled growth chamber room. These precautions help to prevent or reduce water loss from the leaves. In the same line, there are a few studies that have great similarities to our study, and they also used the fresh weights of leaves or diets to check the nutritional indices and growth of *S. frugiperda* [45,60]. Still, future studies should consider the water loss measurement to account for these factors more precisely.

Interestingly, *S. frugiperda* showed elevated larval and pupal performance on castor bean in addition to maize. This can be attributed to matching host plant conditions because of parental pre-exposure to castor bean [61]. Previously, promoted offspring development of *Pieris rapae* and *Coenonympha pamphilus* was reported by eating the same host plants that their parents were fed with [62,63,64]. In another study, the offspring of *Bicyclus anynana* preferred a synthetic odour if their parents were raised on feed treated with that odour [65]. Similarly, a positive influence of the parental diet on the performance of *Spodoptera littoralis* offspring was reported by Rösvik et al. [66]. He clinched that the transgenerational impact of host plants can only affect the progeny’s development in terms of the feed physiology of the offspring, with elevated performance on the given diet, but it cannot change their behaviour. However, such enhanced impacts occur only under favourable conditions. In the same line, Zielonka et al. also proposed that variations in the compositions of the host plant species can alter the offspring performance of polyphagous insect pests [67]. So far, the literature on the transgenerational effects of host plants on the herbivore behaviour, particularly host plant choice, is still lacking.

In multiple studies, maize and castor bean have consistently been reported as the preferred host plants for *S. frugiperda*, which is in line with our results. The ranking of cotton and sugarcane varies, possibly because of geographic variations and differences in methodologies. Our results are slightly different from those of some studies indicating increased *S. frugiperda* damage on cotton in particular areas. The findings of our research generally align with the existing rankings, but they also indicate a decreased preference for cabbage [43]. This indicates that regional factors and plant variations play a significant role in influencing the host plant preferences of *S. frugiperda*.

Despite contradictory studies on the impact of plant nutritional and mineral constituents on insect pests, it is evident that these constituents influence the growth and feeding fitness of herbivorous insects. It is suggested that a sustainable cropping system with a push–pull strategy (non-host and host plants) can be used to reduce pest infestations by repelling the ovipositing herbivores and attracting the pests out of the field to disrupt *S. frugiperda* infestations in maize crop [28]. Also, non-host crops can be attractive to pests and are planted alongside vulnerable plants like maize to attract pests. It will induce oviposition of lepidopteran pests, with reduced larval survival as compared to maize [68,69]. It has also been experienced that using a polyculture cropping system may have reduced pest damage as compared with using a monoculture cropping system in a particular area [70]. Moreover, the nutrient management strategy can also be a part of the cultural control action in IPM to reduce infestations. By focusing on the nutritional and mineral needs of both crops and pests, we can develop integrated approaches that not only improve crop yields but also reduce the impact of insect pests like *S. frugiperda*.

## 5. Conclusions

This study highlights the effect of different host plants, with varied nutrients and mineral contents, on the feeding efficiency, growth, development, and survival of *S. frugiperda*. Maize and castor bean were found to be the most suitable host plants by supporting the highest larval feeding efficiency, growth, development, and survival because of their favourable nutritional and mineral profiles. Keeping in view the nutritional and mineral needs of both crops and pests, strategies like the push–pull technique with a polyculture cropping system and effective nutrient management offer sustainable tactics to reduce the economic impact of this polyphagous pest. Further validation of these findings, through additional research, is necessary to evaluate the chemical and volatile profiles of these cultivated host plants of *S. frugiperda* to use them in a sustainable cropping system.

## Figures and Tables

**Figure 1 insects-15-00789-f001:**
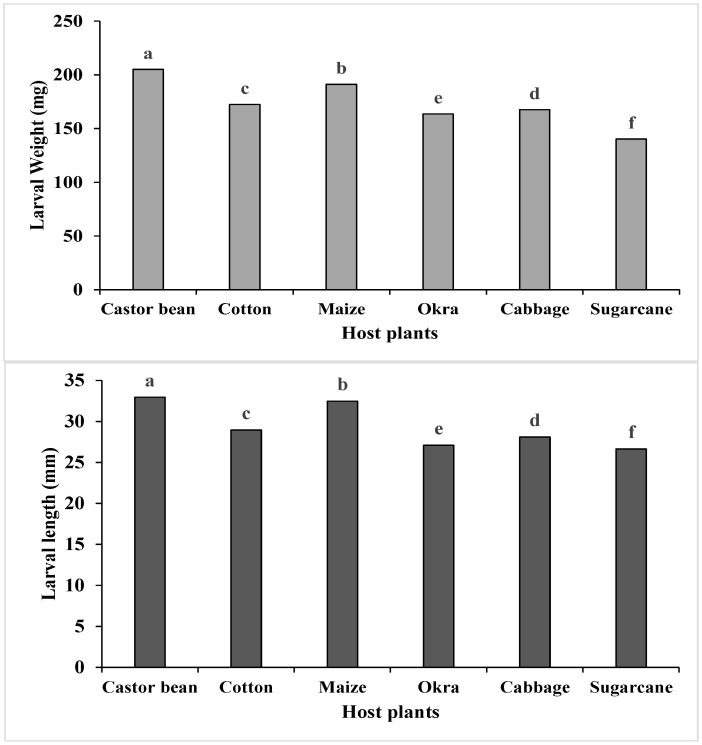
Larval weight (mg ± SE) and larval length (mm ± SE) of *S. frugiperda* in response to their feeding on different host plants (bars for larval weights and larval lengths sharing similar letters are not significantly different from each other according to the Duncan multiple range (DMR) test at *p* < 0.05).

**Figure 2 insects-15-00789-f002:**
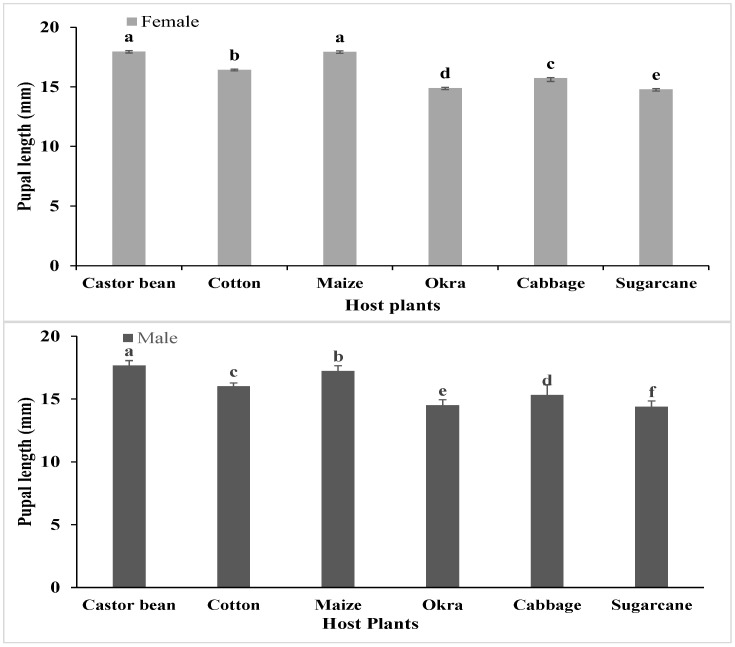
Pupal length (mm ± SE) of *S. frugiperda* in response to their feeding on different host plants (bars for female and male pupal lengths sharing similar letters are not significantly different from each other according to the Duncan multiple range (DMR) test at *p* < 0.05).

**Figure 3 insects-15-00789-f003:**
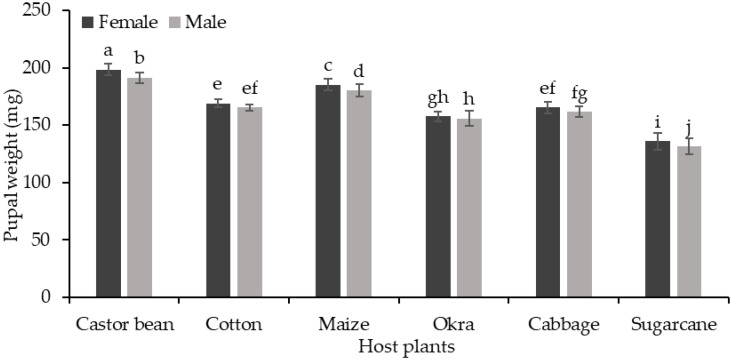
Pupal weight (mg ± SE) of *S. frugiperda* in response to their feeding on different host plants. The interaction was significant (*p* < 0.05) (bars for female and male pupal weights sharing similar letters are not significantly different from each other according to the Duncan multiple range (DMR) test at *p* < 0.05).

**Figure 4 insects-15-00789-f004:**
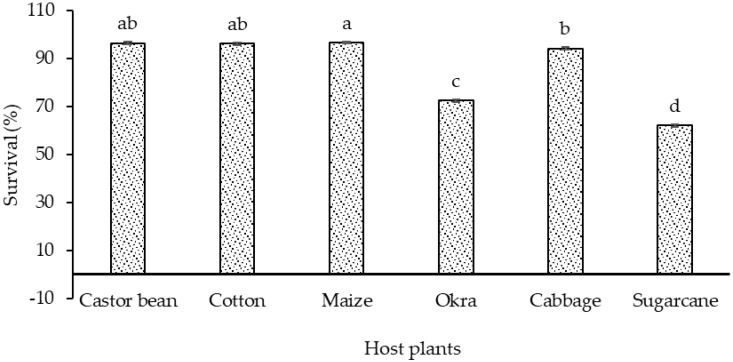
Survival (% ± SE) of *S. frugiperda* larvae in response to their feeding on different host plants (bars sharing similar letters are not significantly different according to the Duncan multiple range (DMR) test at *p* < 0.05).

**Table 1 insects-15-00789-t001:** Proximate composition analysis (% ± SE) of different host plants used as feed for *S. frugiperda* larvae.

Host Plants	Proximate Nutritional Contents				
	Ash	Crude Protein	Crude Fibre	Dry Matter	Crude Fat
Castor bean	10.74 ± 0.13 ^b^	23.48 ± 0.13 ^a^	11.20 ± 0.25 ^d^	43.65 ± 0.15 ^a^	4.05 ± 0.10 ^b^
Cotton	9.17 ± 1.20 ^c^	12.30 ± 0.38 ^d^	9.36 ± 0.32 ^e^	26.47 ± 0.24 ^c^	9.50 ± 0.17 ^a^
Maize	12.13 ± 0.15 ^a^	21.53 ± 0.26 ^b^	28.77 ± 0.33 ^b^	36.17 ± 0.20 ^b^	3.77 ± 0.06 ^bc^
Okra	11.67 ± 0.15 ^a^	11.33 ± 0.20 ^d^	11.13 ± 0.18 ^d^	18.27 ± 0.15 ^e^	3.27 ± 0.1 ^c^
Cabbage	9.45 ± 0.40 ^c^	18.60 ± 0.17 ^c^	21.70 ± 0.17 ^c^	15.80 ± 0.13 ^e^	0.87 ± 0.09 ^e^
Sugarcane	6.67 ± 0.28 ^d^	4.57 ± 0.18 ^e^	33.20 ± 0.15 ^a^	23.24 ± 0.14 ^d^	1.63 ± 0.03 ^d^
Df1	5	5	5	5	5
Df2	12	12	12	12	12
F-value	110.7	1149.4	1742.8	1154.3	292.9
*p*-value	<0.001	<0.001	<0.001	<0.001	<0.001

Means in the columns sharing similar letters are not significantly different according to the Duncan multiple range (DMR) test at *p* > 0.05.

**Table 2 insects-15-00789-t002:** Mineral contents (mg/g ± SE) in the different host plants used as feed for *S. frugiperda* larvae.

Host Plants	Mineral Contents				
	Calcium	Magnesium	Potassium	Phosphorus	Zinc
Castor bean	24.83 ± 0.28 ^b^	3.71 ± 0.10 ^c^	23.96 ± 0.42 ^a^	3.53 ± 0.12 ^c^	0.20 ± 0.03 ^d^
Cotton	2.49 ± 0.17 ^e^	5.70 ± 0.14 ^b^	15.93 ± 0.41 ^c^	11.28 ± 0.32 ^a^	0.36 ± 0.05 ^c^
Maize	6.36 ± 0.14 ^d^	3.53 ± 0.11 ^c^	16.46 ± 0.17 ^c^	1.30 ± 0.04 ^d^	0.63 ± 0.12 ^b^
Okra	28.33 ± 0.26 ^a^	1.53 ± 0.03 ^d^	21.00 ± 0.19 ^b^	11.47 ± 0.27 ^a^	0.82 ± 0.2 ^a^
Cabbage	21.29 ± 0.23 ^c^	38.92 ± 0.3 ^a^	26.31 ± 0.22 ^a^	5.68 ± 0.16 ^b^	0.36 ± 0.04 ^c^
Sugarcane	2.39 ± 0.09 ^e^	1.36 ± 0.03 ^d^	21.02 ± 0.22 ^b^	1.41 ± 0.07 ^d^	0.67 ± 0.13 ^b^
df1	5	5	5	5	5
df2	12	12	12	12	12
F-value	2323.1	2456.8	190.8	99.5	4638.2
*p*-value	<0.001	<0.001	<0.001	<0.001	<0.001

Means in the columns sharing similar letters are not significantly different according to the Duncan multiple range (DMR) test at *p* > 0.05.

**Table 3 insects-15-00789-t003:** Feeding indices of *S. frugiperda* larvae in response to their feeding on different host plants.

Host Plants	RGR	RCR	ECI	CI
Castor bean	0.97 ± 0.03 ^b^	47.13 ± 0.45 ^b^	71.80 ± 1.25 ^a^	67.68 ^ab^
Cotton	0.85 ± 0.01 ^c^	33.49 ± 0.47 ^c^	71.11 ± 1.32 ^a^	67.84 ^ab^
Maize	1.13 ± 0.03 ^a^	50.47 ± 0.61 ^a^	70.77 ± 1.45 ^a^	68.39 ^a^
Okra	0.71 ± 0.03 ^d^	19.66 ± 0.46 ^d^	69.22 ± 2.03 ^a^	67.08 ^ab^
Cabbage	0.77 ± 0.01 ^cd^	33.22 ± 0.50 ^c^	69.87 ± 1.40 ^a^	67.47 ^ab^
Sugarcane	0.59 ± 0.01 ^e^	19.2 ± 0.40 ^d^	70.02 ± 1.69 ^a^	65.69 ^b^
df1	5	5	5	5
df2	20	20	20	20
F-value	41.5	486.6	2.61	3.14
*p*-value	<0.001	<0.001	0.066	0.03

Means in columns sharing similar letters are not significantly different according to the Duncan multiple range (DMR) test at *p* > 0.05.

**Table 4 insects-15-00789-t004:** Matrix of Pearson’s correlation coefficient among proximate analysis and feeding indices of *S. frugiperda*.

	RGR	RCR	DM	CP	CF	EE	Ash
RCR	0.91 **						
DM	0.50 **	0.64 **					
CP	0.21	0.23	0.43 *				
CF	−0.10	−0.05	−0.30	−0.73 **			
EE	0.29	0.21	0.13	−0.04	−0.59		
Ash	0.69 **	0.53 **	0.26	0.62 **	−0.38	0.10	
ECI	−0.30	0.43	0.01	0.13	0.64	0.52	0.26

RGR, relative growth rate; RCR, relative consumption rate; ECI, efficiency of conversion of ingested food; DM, dry matter; CP, crude protein; CF, crude fibre; EE, ether extract/crude fat. * indicates significant correlation and ** indicates highly significant correlation.

## Data Availability

The data presented in this study are available on request from the first corresponding author.

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
