# Peer review of "Feeding and Growth Response of Fall Armyworm Spodoptera frugiperda (Lepidoptera: Noctuidae) towards Different Host Plants"

_insects, 2024, doi:10.3390/insects15100789_

Round 1

Reviewer 1 Report

Comments and Suggestions for Authors

I have attached my suggestion as a pdf file. 

Comments on the Quality of English Language

I recommend native speakers edit the English language of the manuscript.  

Reviewer 2 Report

Comments and Suggestions for Authors

Ajmal et al. report the results of a laboratory study on the effect of feeding different host plants to FAW larvae.
The manuscript has important deficiencies.
MAIN ISSUES
I) The manuscript is poorly written and difficult to understand and repetitive in parts.
II) I suspected that the authors had used ChatGPT in some parts of their Introduction and Discussion, but they do not report the use of AI tools as specified in MDPI guidelines.
III) The use of "fitness indices" seems unnecessary and confusing to me. For example, the larval growth index is based on the duration of the larval development period and the percentage of larvae that pupate. These are two distinct response variables that should be considered separately (as in the huge majority of entomological studies).  There is no benefit in combining them into an "invented" and uninformative "growth index". All this section should be removed from the manuscript for improved clarity.
IV) Some of the data appear to have issues of heteroscedasticity and non-normality that would invalidate the Analyses of Variance reported in the Results section. The authors have not reported the error degrees of freedom for their F statistics, so I was uncertain of other aspects of their analyses.

I have written numbered points and suggestions on a scanned copy of the manuscript.
Numbered points (see scanned file)
1. The term "fitness response" is misleading or confusing. You measured growth and feeding. You did not measure reproduction, so all responses were indirectly related to fitness. Suggest you delete fitness from the title.
2. What is feeding fitness? Does such a concept exist?
3. Better nutrients? What does this mean? Reword.
4. Reword.
5. You mean in larvae that fed on the same host plants.
6. State what you mean by proximate compositions for clarity (crude fiber, ash, etc.)
7. p<0.5 is NOT significant.
8. Do not use title words as keywords.
9a. From a reading of the Introduction, I suspect that you used ChatGPT or similar tools. The text is highly repetitive and has many errors.
9b. Chemical defenses are not physical barriers. Reword.
10. Obvious, trivial text. Delete.
11. You start by talking about mineral content then switch to nutrient content. Please be consistent in the subject of each sentence.
12. If specific host plant resistance reduces crop losses, why would you include a high species richness that could be more susceptible to attack?  Reword.
13. This text is repetitive.
14. You outline the importance of mineral content at different points. This needs to be reorganized into coherent sections.
15. This text is repetitive.
16. This text is incoherent. Written by AI?
17. The first 5 paragraphs need to be reduced in length, organized coherently and focused on the main issues of the study.
18. The detailed feeding behavior of FAW on maize is not the focus of the study. Delete.
19. What value is this text? Suggest you delete this paragraph.
20. Why did you select these as host plants for study? You state in L101 that the greatest threat of FAW is to maize followed by rice, but rice was not included in your study.
21a. State why you reared FAW larvae on castor bean (an uncommon host) rather than maize. What are the consequences of this decision on the outcome of the study?
Section 2.3 What were the varieties of each of the crop plants used in the study?
21b. Mention specifically what you mean by proximate composition.
22. blinder? Do you mean a blender?
23. How many replicates were performed?
How many plants were sampled? How many leaves from each plant?
How were the leaves selected?
What was the sample size for each analysis?
24a. You mention insect length in the Results but not in the Methods (section 2.5).
24b. It is unclear how often you measured diet/feces/larval wt - - at intervals of 24h? Clarify and reword.
24c. What was the sample size here? Did you measure all 50 larvae from each group?
24d. How was insect length measured? Larvae are highly active and do not stay still - did you use CO2?
25a. Are all these wet weights? What about water loss during the experiment?
25b. L187 says diet weight? But "diet" (artificial diet) was not used - you mean leaf weight?
25c. Leaves of different plants lose water at very different rates over a 24 h period. Maize is highly susceptible to water loss whereas castor bean and cabbage less so.  How did you account for this?
26. Here you are talking about dry weights (biomass) correct?  Please clarify this in the text.
27. The larval period and the percentage of pupation seem to be unrelated. Why do you consider this to be a useful index? I have not seen this used elsewhere (except in a previous study by the same authors). The same applies to the other "indices" which seem to have little value compared to the original variables (emergence, weight, developmental period).
28. How did you ensure that your data met the equality of variances assumption of ANOVA?
The same applies to the normality of the data. It is clear in the Results that you faced issues that could have invalidated ANOVA based analyses.
29. What do letters indicate in Fig 1, Fig 2? Comparisons within host plants? or among host plants?
30. P >0.05 is not significant.
31. You clearly have normality and heteroscedasticity issues in the data in Fig 2.
32. F statistics are ratios derived from treatment variation and error (residual) variation. F stats can only be understood if you report the treatment (df1) and error (df2) degrees of freedom.
33. If F 3.45 gives a p value of 0.08, how does F of 3.14 give a much smaller p value with the same degrees of freedom. There seems to be an error.
34. What do letters indicate in Fig 3?, Fig 4, Fig 5?
35. Please report percentages to one decimal place, as you only had modest sample sizes and 5 replicates.
36. Please delete these "indices" which are not informative in my opinion and seem to have no biological relevance.
37. Please rationalize this text and Table 3, Table 4 by removing the unnecessary "indices". You should retain the analysis of the individual response variables (development time, wt, etc.)
38. Delete this repetitive text (already stated in the Introduction).
39. Delete obvious text.
40. climaxes? You mean highlights?
41. In the entire Discussion you do not consider the effect of prior insect diet on the response to each of the host plants. There are several studies that consider this issue - especially when using artificial diets in Sf and other noctuids.
42. Gypsy moth is now considered as a racial slur and has been renamed "spongy moth".
43. Females are often heavier than males because fitness gains are accrued at a higher rate through  increased body weight in females compared to males. There is a lot of literature on this in lepidopterans.
44. Why would you suggest planting host plants (albeit non-preferred) in mixed cropping systems? This would provide more food for FAW. Surely the recommendation would be to use NON_host plants to repel FAW (e.g. https://doi.org/10.7554/eLife.88695 ) in a push-pull strategy.
45. This is generic text, not a useful conclusion. Please focus on your main findings.
46. The final sentence is vague and needs rewording.
How do you rank the host plants in order of value to FAW and how does your ranking compare to that of other authors? (there are MANY studies from the Americas, and Asia on this topic).

Comments on the Quality of English Language

Needs editing.

Round 2

Reviewer 1 Report

Comments and Suggestions for Authors

The author has made significant changes to the manuscript, which is a positive step. However, there are still some minor points that need to be revised. In addition, some of my suggestions have been deleted from the revised version of the text, which looks OK. Still, the authors mentioned in the 'answer to reviewer' letter that those suggestions have been added or done. It was confusing because I could not find the words or sentences the author mentioned 'added' or 'corrected'! They should write that those suggestions were deleted from the text in the revised version. 

Here are some points that still need revision.

1- Please provide a degree of freedom (df) inside the parenthesis for all of the statistical analysis in the text. 

2- please use a white background for all of the figures.

3- please correct the sentence in the material and method to this:" The city is located in a plain area at an altitude of 214 meters above sea level, with a longitude of 29° 25' 5.0448'' N and a latitude of 71° 40' 14.4660'' E."

Reviewer 2 Report

Comments and Suggestions for Authors

The authors have addressed most of the editorial suggestions that I pointed out previously. However two major statistical issues remain and two additional issues:

(I) As previously stated, some of the data appear to have issues of heteroscedasticity (inequality of variances) that would invalidate the Analyses of Variance reported in the Results section. The authors state that they have now transformed some of their data to obtain normally-distributed data, but they have NOT addressed heteroscedasticity. The authors may be unaware that violating assumptions of equality of variances has a more profound effect on the reliability of ANOVA analyses than non-normality (e.g. doi: 10.1136/gpsych-2019-100148).

(II) As I previously stated, F statistics are ratios derived from treatment variation and error (residual) variation. F stats can only be understood if you report the treatment (df1) and error (df2) degrees of freedom. The authors have now provided treatment degrees of freedom but have neglected to provide error (residual) degrees of freedom for the F statistics. This is of crucial importance as the error df indicate the sample size and reveal possible issues of pseudoreplication. As the SE values given in the figures are in all cases tiny, I was concerned that the authors may have analyzed individual insects as independent observations (which they are not!). The only way to determine the correct analysis of these data is by examining the error df values of the F statistics. Error df values are missing in Table 1, Table 2, Table 3, and in the text on lines 215, 216, 183, 232, 242, 246.

(III) The issue of water loss from each of the different types of leaves has not been addressed in the Discussion. The authors state that they did their best to reduce water loss during the experimental period, but they did not MEASURE water loss. The different types of leaves (e.g. cotton vs. cabbage) differ markedly in their tendency to lose water. The authors need to be up front about this issue in their study.

(IV) In the Discussion, the authors cite literature that does not consider S. frugiperda, but this is not made clear in the Discussion. For example, [43] was performed on a beetle, [52] was performed on a fly, [53] was performed on a different Spodoptera species (S. littoralis), and when mentioning effects on pupae the authors cite a study on an aphid (which does not have a pupal stage!). The authors need to revise their citation of the literature in the entire manuscript to make it clear when they are referring to S. frugiperda as opposed to other species of insects.

Comments on the Quality of English Language

Moderate editing required.

Round 3

Reviewer 2 Report

Comments and Suggestions for Authors

The authors have made modifications, but they continue to struggle with reporting their statistical inferences correctly.

1. P values generated from Levene's test should not be reported in Tables 1, 2 or 3. This generated confusion between Levene's test P values and F statistic (ANOVA generated) P values.

2. F statistics are conventionally reported in the following form:  F = 1.23; d.f. = 5, 20; P = 0.647 and need to be corrected in the entire manuscript.

3. The P values shown in line 186 and 226 are incorrect and should be P >0.05 (not P < 0.05).

4. Section 2.7 should mention the use of Levene's test to check for equality of variances.

5. The authors need to correct the term "leave" (which means going away) to leaf (singular) or leaves (plural) on lines 265, 365, 369.

6. Once again the authors have omitted to report the error degrees of freedom for F statistics on lines 247, 257, 261, 276. This is the third time that I have requested this.

Comments on the Quality of English Language

Minor editing required.
